# OPTIMIZING DATA-FLOW IN BINARY NEURAL NETWORKS

## ABSTRACT

Binary Neural Networks (BNNs) can significantly accelerate the inference time of a neural network by replacing its expensive floating-point arithmetic with bit-wise operations. Most existing solutions, however, do not fully optimize data flow through the BNN layers, and intermediate conversions from 1 to 16/32 bits often further hinder efficiency. We propose a novel training scheme that can increase data flow and parallelism in the BNN pipeline; specifically, we introduce a clipping block that decreases the data-width from 32 bits to 8. Furthermore, we reduce the internal accumulator size of a binary layer, usually kept using 32-bit to prevent data overflow without losing accuracy. Additionally, we provide an optimization of the Batch Normalization layer that both reduces latency and simplifies deployment. Finally, we present an optimized implementation of the Binary Direct Convolution for ARM instruction sets. Our experiments show a consistent improvement of the inference speed (up to $1.77$ and $1.9\times$ compared to two state-of-the-art BNNs frameworks) with no drop in accuracy for at least one full-precision model.

## 1 INTRODUCTION

In the last decade deep neural networks (DNNs) have come to demonstrate high accuracy on many datasets like ImageNet Russakovsky et al. (2015), outperforming legacy methods and sometimes even human experts(Krizhevsky et al. (2012), Simonyan & Zisserman (2014), Szegedy et al. (2015), He et al. (2016)). These improvements have been achieved by increasing the depth and complexity of the network, leading to intensive usage of computational resources and memory bandwidth. Large DNN models run smoothly on expensive GPU-based machines but cannot be easily deployed to edge devices (i.e., small mobile or IoT systems), which are typically more resource-constrained. Various techniques have been introduced to mitigate this problem, including network quantization Choi et al. (2018); Hubara et al. (2016); Lin et al. (2017); Rastegari et al. (2016); Zhou et al. (2016), network pruning Han et al. (2015); Wen et al. (2016) and efficient architecture design Howard et al. (2017); Tan & Le (2019).

Recent work on quantization (e.g. Courbariaux et al. (2016); Hubara et al. (2016); Liu et al. (2018); Martinez et al. (2020)) has shown that a DNN model can be even quantized to 1-bit (also known as binarization) thus achieving a remarkable speedup compared to the full precision network. The memory requirement of such a binarized DNN (BNN) is drastically reduced compared to a DNN of the same structure, since a significant proportion of weights and activations can be represented by 1-bit, usually $\{-1, +1\}$. In addition, high-precision multiply-and-accumulate operations can be replaced by faster XNOR and popcount operations.

However, the aggressive quantization can make BNN's less accurate than their full-precision counterparts. Some researchers showed that the performance loss often arises from the gradient mismatch problem caused by the non-differentiable binary activation function Darabi et al. (2018); Liu et al. (2018). This non-differentiability of the quantization functions prevents gradient back-propagation through the quantization layer. Therefore, previous works used straight-through-estimator (STE) to approximate the gradient on non-differentiable layers Bengio et al. (2013); Hubara et al. (2016).

Furthermore, to prevent that the binarization of weights and activations leads to feature maps of lower quality and capacity, a combination of binary and floating-point layers is usually adopted. Unfortunately, each time a binary layer is connected to a floating-point one, the efficiency of the

pipeline is compromised by input/output layer data type conversion. In addition, the internal parallelism of a binary layer depends on the encoding of the accumulator, which is often maintained at 32 bits to prevent overflow. In this paper we present several optimizations that allow training a BNN with an inter-layer data width of 8 bits. Most prior work on BNN's emphasize overall network accuracy; in contrast, our aim is to preserve initial accuracy while improving efficiency. Our contributions (graphically highlighted in Figure 1i and 1ii) can be summarized as follows:

- a novel training scheme is proposed to improve the data-flow in the BNN pipeline (Section 3.1); specifically, we introduce a clipping block to shrink the data width from 32 to 8 bits while simultaneously reducing the internal accumulator size.
- we provide (Section 3.2) an optimization of the Batch Normalization layer that decreases latency and simplifies deployment.
- we optimize the Binary Direct Convolution method for ARM instruction sets (Section 3.3).

To prove the effectiveness of the proposed optimizations in Section 4 we provide experimental evaluations that show's the speed-up relative to state-of-the-art BNN engines like LCE Bannink et al. (2021) and DaBNN Zhang et al. (2019).

## 2 RELATED WORK

BNNs were first introduced by Courbariaux et al. (2016), who established an end-to-end gradient back-propagation framework for training the binary weights and activations. They achieved good success on small classification datasets including CIFAR10 Krizhevsky et al. (2009) and MNIST Netzer et al. (2011), but encountered a severe accuracy drop on ImageNet.

Many subsequent studies focused on enhancing BNN accuracy. Rastegari et al. (2016) proposed XNOR-Net, where real-valued scaling factors are used to multiply the binary weight kernels, and this methodology then became a representative binarization approach to bridge the gap between BNN's and their real-valued counterparts. The Bi-Real Net Liu et al. (2018) added shortcuts to propagate values along the feature maps, which further boosted the top-1 accuracy on ImageNet; nevertheless, the model still relies on 32-bit floating point to execute batch normalization and addition operator (as shown in Fig. 1iia).

One of the major weaknesses of BNN's is the gradient approximation by the STE binarization function Courbariaux et al. (2016). In fact, STE computes the derivative of sign as if the binary operation was a linear function, as reported in the following formula :

$A(x) = \max(-1, \min(1, x)), \ STE(x) = A'(x) = [-1 \le x \le 1]$

The implementation of STE stated above, uses the STE with the addition that it cancels the gradients when the inputs get too large Hubara et al. (2016). STE provides a coarse approximation of the gradient that inevitably affects the testing accuracy of the BNN. To address this issue, other recent studies tried to improve the performance of BNNs by adopting a proper optimization method for the quantization. Inspired by STE, many works update the parameters approximately introducing auxiliary loss functions Gu et al. (2019); Qin et al. (2020).

Besides many efforts to develop more efficient and accurate architectures, a few works have provided benchmarks on real devices such as ARM processors. Based on the analysis provided in Bannink et al. (2021), the fastest inference engines for binary neural networks, with proven benchmarks (Section 4 of Bannink et al. (2021)), are LCE and DaBNN.

## 3 DATA-FLOW OPTIMIZATIONS

As illustrated in Fig. 1a, the most commonly used BNN architectures (e.g., VGG and ResNet) have four essential blocks in each convolution/fully-connected (CONV/FC) layer: sign (binarization), XNOR, popcount and Batch Normalization (BN). Since the weights, inputs and outputs are all binary, the traditional multiply-and-accumulate operation is replaced by XNOR and bit counting (i.e., popcount). XNOR and popcount are usually fused to improve efficiency. The use of Batch Normalization after each binarized layer is very important in BNN's as pointed out by Santurkar et al.

(2018). Figures 1i and 1ii (b and c) point out our proposed BNN optimizations during training and inference. Before discussing them in detail, we show the data-flow bottlenecks that affect existing solutions and then describe how to reduce them.

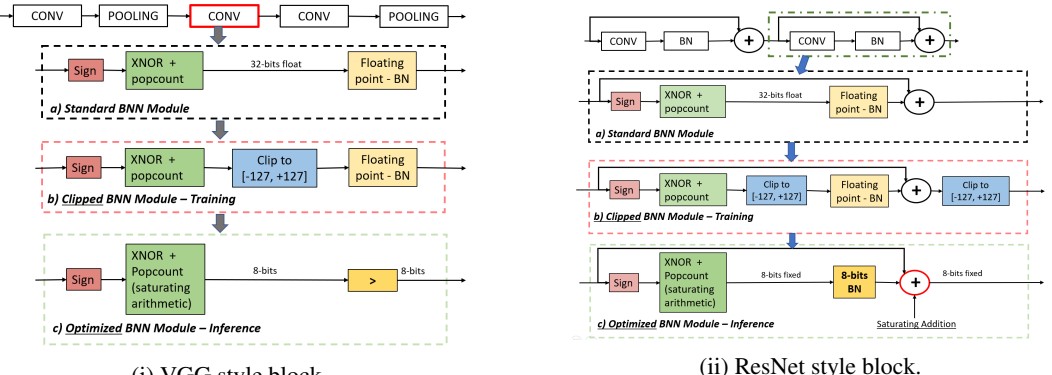

| (i) VGG style block. | (ii) ResNet style block. |

Figure 1: a) Standard BNN blocks used in Rastegari et al. (2016) and Liu et al. (2018). b) BNN block with output convolution clipping used during training. c) Optimized BNN block adopted during inference. Popcount operation is performed using saturation arithmetic in order to keep the data width to 8 bits at inference time. BN is replaced in by a comparison in case i, while in ii BN is 8-bit quantized.

In Figs. 2i and 2ii we report an example of binary convolutional layer outputs for a VGG and a ResNet model. The ranges of activation values after popcount (green histograms) exceed the interval $[-128; +127]$ [1], so adopting an 8-bit encoding would lead to overflow. To prevent such a data loss, most of the existing BNN frameworks (including Bannink et al. (2021); Zhang et al. (2019)) encode such data in 32-bit floating point. On the other hand, the ranges of values after BN (red histograms in Fig. 2) are more limited.

In this paper, we propose to quantize the popcount output with 8-bit integers through a two-stages training procedure, which is designed to preserve model accuracy. In the next subsection we show how to apply this technique to VGG and ResNet models.

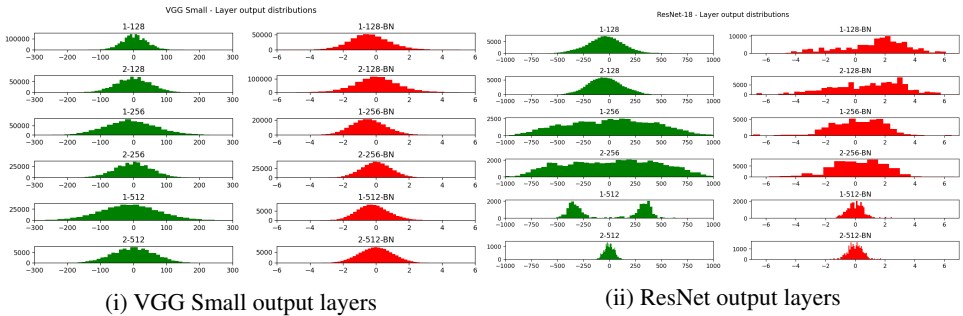

| (i) VGG Small output layers | (ii) ResNet output layers |

Figure 2: Example of output distributions after binary convolution. i, refers to a VGG style network while ii to a ResNet architecture. Green shows the distribution before the BN layer and red afterward.

## 3.1 TWO-STAGE CLIPPING

Our training procedure selectively executes or skips a clipping operation at each binary layer (row *b* of Figs. 1i and 1ii, blue blocks). A two-stage training method is introduced to avoid accuracy loss when clipping is enabled: during a first warm-up stage, the model is trained without any range constraints, while in the second stage (details are reported in Algorithm 1) the network is trained with

---

[1] We actually consider the symmetric quantization interval $[-127; +127]$ because this choice enables a substantial optimization opportunity, as reported in Appendix B of Jacob et al. (2018).

the clipping block enabled. Based on the high accuracy reached at the end of the first training stage, in the second training stage the model better tolerates clipping 8-bit quantization; we experimentally found that this approach preserves the accuracy of a model that does not contain clipping.

---

**Algorithm 1:** Second stage training procedure for BNNs

**Input:** The full-precision weights W; the input training dataset
**Output:** BNN model with convolution output clipped

1   Initialize network weights $W$
2   **repeat**
     // Forward Propagation
3     **for** $l = 1$ to $L$ **do**
4       Binarize floating point weights: $W_{bin}^l = sign\left(W^l\right)$
5       Binarize floating point features of previous layer: $F_{bin}^{l-1} = sign\left(F^{l-1}\right)$
6       Compute binary convolutions features: $F_{out}^l = F_{bin}^{l-1} * W_{bin}^l$
7       Clip $F_{out}^l$ values to interval $[-127; +127]$ with:
       $F_{out\,clipped}^l = max\left(min\left(127, F_{out}^l\right), -127\right)$
8       Perform Batch Normalization: $BN\left(F_{out\,clipped}^l\right) = \gamma^l \frac{F_{out\,clipped}^l - \mu^l}{\sigma^l} + \beta^l$
     // Backward Propagation
9     **for** $l = 1$ to $L$ **do**
10      Compute gradients based on the binarization weights $W_{bin}^l$, clipped convolutions
       $F_{out\,clipped}^l$ and batch normalization output $BN\left(F_{out\,clipped}^l\right)$
11      Update full-precision weights $W^l$
12   **until** *convergence*

---

### 3.2 BATCH NORMALIZATION OPTIMIZATION

The BN layers after the clipping are also optimized/8-bit quantized to further increase the data-flow of the inference pipeline. The Batch Normalization layer scales and shifts the output of the CONV/FC layer as follows:

$$BN\left(F_{out}^l\right) = \gamma \frac{F_{out}^l - \mu}{\sigma} + \beta \tag{1}$$

where $\gamma$, $\mu$, $\sigma$ and $\beta$ are learned parameters and $F_{out}^l$ is the ouput feature of layer $l$ that is the input of BN function.

The BN optimization depends on the network model: VGG or ResNet. In both cases we show that it is possible to keep the inter-layer data type to 8-bit with appropriate changes to the binary layer structure.

- **VGG style block**. When the BN layer is inserted in a pipeline similar to Fig. 1i, where the following block is still binary, the BN operation can be simplified replacing multiplication and division in Eq. 1 with a simple comparison with a threshold $\tau$. The simplification of Eq. 1 leads to:

$$sign\left(BN\left(F_{out}^l\right)\right) = \begin{cases} +1 & if\ BN\left(F_{out}^l\right) \geq 0 \\ -1 & otherwise \end{cases}$$

$$\gamma \frac{F_{out}^l - \mu}{\sigma} + \beta \geq 0 \Rightarrow \tau \doteq \mu - \beta \frac{\sigma}{\gamma} \tag{2}$$

$$sign\left(BN\left(F_{out}^l\right)\right) = \begin{cases} +1 & if\ F_{out}^l \geq \tau\ else\ -1 & \left(when\ \frac{\gamma}{\sigma} \geq 0\right) \\ -1 & if\ F_{out}^l \leq \tau\ else\ +1 & \left(when\ \frac{\gamma}{\sigma} < 0\right) \end{cases}$$

The threshold $\tau$ of Eq. 2 can be computed offline and easily quantized to 8 bits in order to exploit the output features of layer $l$. Therefore, when multiple BNN modules are stacked, Batch Normalization can be replaced by a threshold comparison according to Eq. 2.

- **ResNet style block**. When a BNN block is placed in a ResNet style pipeline, followed by an addition operator, Fig. 1ii, the BN layer can be executed while both scaling and bias factors to 8 bits. As reported in Fig. 3, the internal data representation of a quantized BN layer is expanded to 16-bit to preserve accuracy during quantization but the input/output data type still remains within 8 bits. The complete iterative quantization procedure we adopted is reported in Algorithm 2 of Appendix 6. The procedure iterates over floating-point layers (basically all BN layers) inside the binary blocks and, for each one: computes the quantization scale; quantizes; freezes the weights; and retrains the remaining layers.

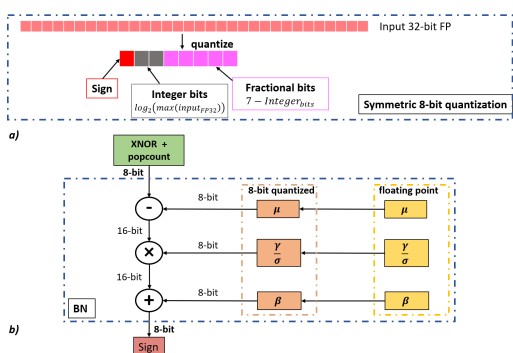

Figure 3: **a**): 8-bit symmetric quantization procedure that reserves fractional/integer bits based on the range of input 32-bit floating point values. **b**): implementation of the BN layer with 8-bit quantization using an internal 16-bit representation to preserve accuracy.

### 3.3 BINARY DIRECT CONVOLUTION OPTIMIZATION ON ARM

The GEMM (GEneral Matrix Multiplication) is a widely adopted method to efficiently implement convolutions. However, as reported in Zhang et al. (2018b), the GEMM approach increases the memory footprint of the model, making a model's port to an embedded device more difficult. Furthermore, GEMM routines are not always optimized for convolutions on ARM devices, in particular ARMv8 (64-bit ARM architecture) and its relevant operations such as *vcount* and *addv*.

*vcount* takes an N-byte vector as input and outputs an N-byte vector containing the number of $1s$ present in each input byte. *addv* takes an N-byte vector as input and outputs the sum of the N bytes as one single value.

Inspired by Zhang et al. (2018b) and Zhang et al. (2019) we propose a hybrid direct binary convolution (see Fig. 4) that uses both the *addv* instruction and the common *add* operations. The binary convolution is usually composed of three different steps: binarization/bit-packing, padding and convolution. Zhang et al. (2019), executes these steps in a sequential way. In contrast, we devise a more cache-friendly approach that collapses the previous steps in one operation executed with tiling. We also devise a different kernel memory layout that better fits ARMv8 SIMD processing instructions, as illustrated in Fig. 4.

The implementation details of our binary convolution are reported in Fig. 5. The operation *Extract sign bit* executes the binarization, bit-packing and padding. Then, the (bit-wise) XNOR output is consumed by the popcount operation (*vcnt 8-bit wise*, *add* and *addv*). On the ARM architecture, the latter can be implemented with *vcount* and a sequence of additions (*addv* instructions). We decided to implement several pair-wise additions and only a final *addv* instruction (which is more expensive). The entire convolution process does not provide intermediate outputs but instead processes the input data as a whole. It is worth noting that the clipping operation can be obtained for free on ARM devices by exploiting its saturation arithmetic; all the addition operations (*add* and *addv*) can be limited to the fixed range $[-127; +127]$ by simply adding the postfix $q$ to the operations and executing $\max$ to avoid $-128$ value.

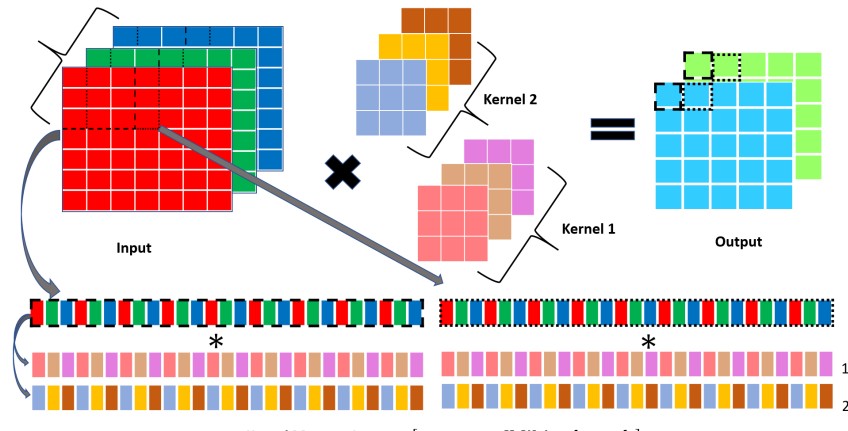

Figure 4: The $7 \times 7$ input image with 3 different channels (denoted by color) is convolved with two separate kernels to obtain a $5 \times 5$ output with two output channels. To better exploit the SIMD 128-bit registers a different memory layout for kernel is devised: $[out_{channels}, H_{filter}, W_{filter}, in_{channels}]$.

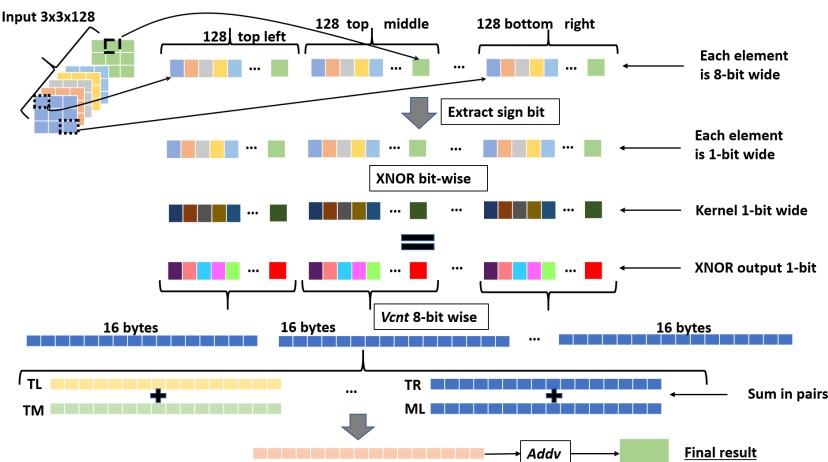

Figure 5: The $3 \times 3 \times 128$ input patch is convolved (XNOR + popcount) with one kernel through the Extract sign bit, XNOR and then popcount operations. Popcount is performed using *vcnt*, summing in pairs the *vcnt* output and the last step uses the *addv* operation. TL (top left), TM (top middle), TR (top right) and ML (middle left) indicate the position of elements inside the $3 \times 3$ patch.

## 4 EXPERIMENTAL RESULTS

In this section, we first evaluate the efficiency result of our approach compared to the state-of-art BNN frameworks such as LCE and DaBNN; the comparison is carried out on real hardware devices like Raspberry Pi Model 3B and 4B with 64-bit OS. Then, we present various accuracy benchmarks of the proposed two-stage training procedure focusing on CIFAR-10, SVHN and ImageNet, and to two different architectures: VGG and Resnet-18.

### 4.1 EFFICIENCY ANALYSIS

To validate the efficiency of our method we focused on the convolution macro-block of Fig. 1 and compared the efficiency of the proposed approach with LCE and DaBNN, which, to the best of our knowledge, are the fastest inference engines for binary neural networks.

Our assessment was performed on ARMv8 platforms, Raspberry Pi 3B and 4B. We implemented, differently from our predecessors, the convolution operation using ARM NEON *intrinsics* instead of inline assembly. Intrisics allow to produce code easier to maintain and automatically fit both ARMv7 and ARMv8 platforms without losing appreciable performance compared to pure assembly code. In Fig. 6 we compare implementations on targets Rpi 3B and 4B. Our solution shows a clear performance improvement for single binarized convolutions for all kernels; the performance boost is more evident when the binarization/bitpacking and data conversion operations are considered in addition to binary convolution only. In summary, our solution, including all the optimizations introduced in Section 3, accelerates binary convolution up to 1.77 and 1.9× compared to LCE and DaBNN.

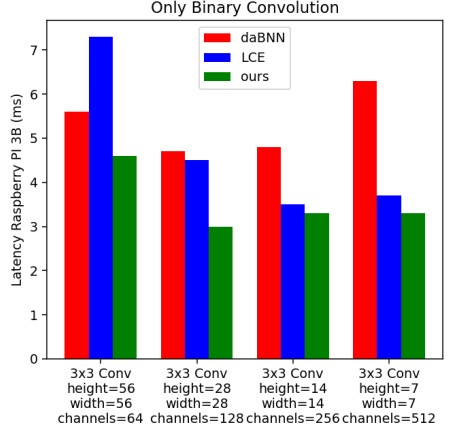 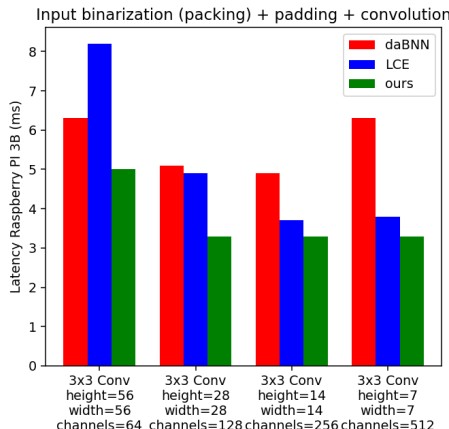

(i) Raspberry Pi 3 benchmark.

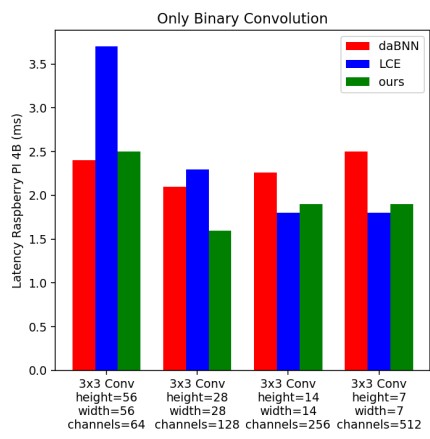 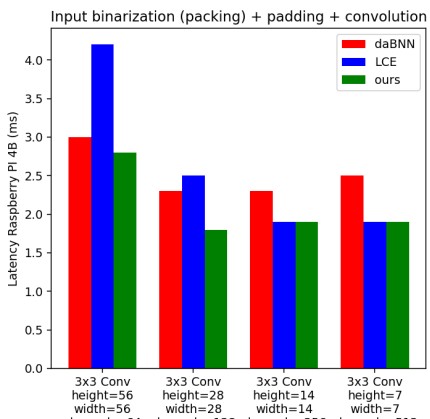

(ii) Raspberry Pi 4 benchmark.

Figure 6: Latency evaluation of our method compared to DaBNN and LCE on Raspberry Pi 3B (i) and 4B (ii) devices. The improved lower latency of our approach is accentuated when considering not only the binary convolution itself (left) but bitpacking and padding as well (right).

## 4.2 ACCURACY ANALYSIS

We evaluated two VGG-style networks for CIFAR-10 and SVHN: VGG-11 Xu et al. (2019) and VGG-Small Zhang et al. (2018a) which are both high-capacity networks for classification. Pre-trained Larq binary models (BinaryResNetE18 and BinaryDenseNet28) were adopted to evaluate the accuracy on ImageNet.

**Results on CIFAR10 and SVHN.** For CIFAR10 the RGB images are scaled to the interval $[-1.0; +1.0]$ and the following data augmentation was used: zero padding of 4 pixels for each

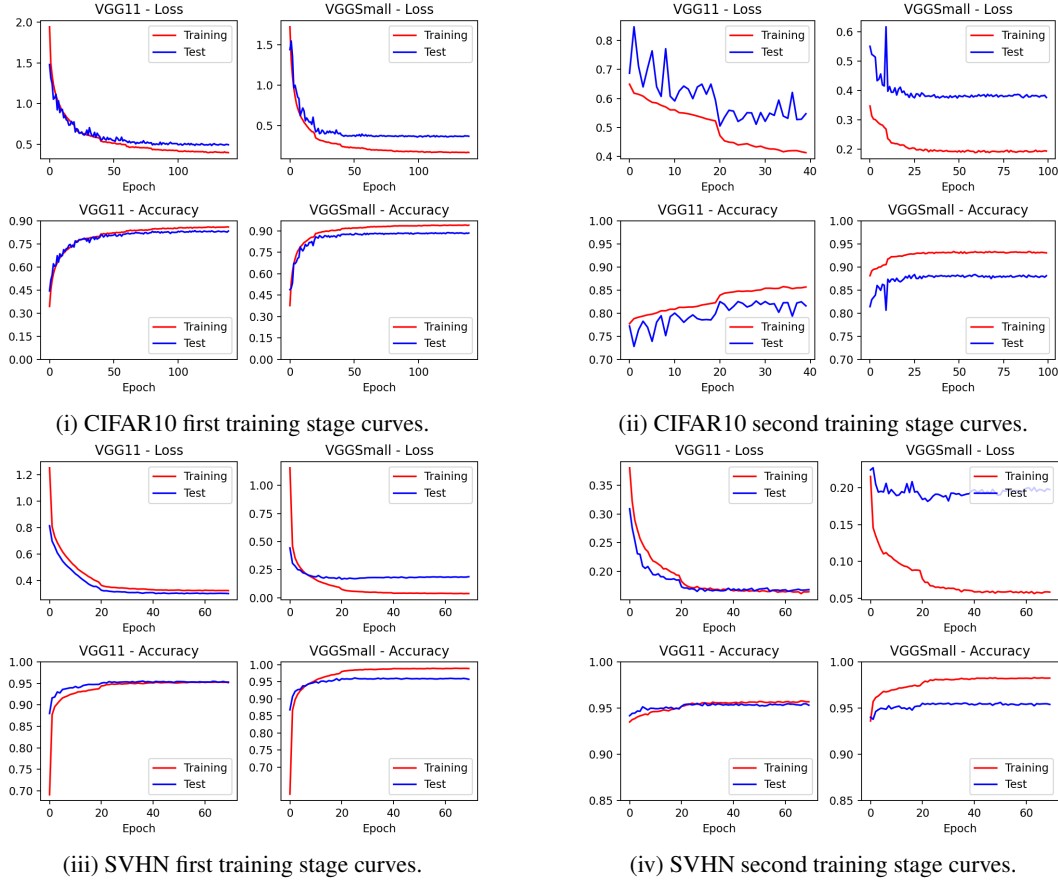

(i) CIFAR10 first training stage curves.

(ii) CIFAR10 second training stage curves.

(iii) SVHN first training stage curves.

(iv) SVHN second training stage curves.

Figure 7: Training loss and testing accuracy curves for VGG11 and VGGSmall on CIFAR10 and SVHN of the first and second training stages.

side, a random $32 \times 32$ crop and a random horizontal flip. No augmentation is used at test time. The models have been trained for 140 epochs.

On SVHN the input images are scaled to the interval $[-1.0; +1.0]$ and the following data augmentation procedure is used: random rotation ($\pm 8$ degrees), zoom ($[0.95, 1.05]$), random shift ($[0; 10]$) and random shear ($[0; 0.15]$). The models have been trained for 70 epochs.

The accuracy achieved by the models is reported in Table 1 showing that the clip operation does not substantially affect the overall accuracy and the two-stage clipping allows to preserve the original accuracy. Fig. 7 shows the training and validation curves on CIFAR10 and SVHN; we can note that a limited number of epochs is necessary during the second training stage.

| Method | Topology | Bit-width | CIFAR10 top1 % | SVHN top1 % |
|---|---|---|---|---|
| BNN Courbariaux et al. (2016) | VGGSmall Zhang et al. (2018a) | 32 FP | 93.8 | 96.5 |
| Main/Subsidiary Network | VGG11 Xu et al. (2019) | 32 FP | 83.8 | - |
| BNN | VGGSmall | 1-bit | 89.9 | 96.5 |
| XNOR-Net Rastegari et al. (2016) | VGGSmall | 1-bit | 82.0 | 96.5 |
| Bop Helwegen et al. (2019) | VGGSmall | 1-bit | 91.3 | - |
| Main/Subsidiary Network | VGG11 | 1-bit | 82.0 | - |
| **ours** | VGGSmall | 1-bit | 88.8 | 96.1 |
| **ours** | VGG11 | 1-bit | **83.7** | 95.5 |

Table 1: Accuracy comparison of our method with SOTA on CIFAR10 and SVHN.

**Results on ImageNet**. Tests were performed by using pre-trained binary versions of ResNet18 and DenseNet28 Bethge et al. (2019) taken from *zoo literature of Plumerai* Larq. Each BNN module (refer to Fig. 1) has been modified according to Fig. 1ii. Residual blocks seem to be more robust to clipping compared to VGG style blocks (Results are in Table 2).

| Method | Topology | Bit-width | top1 % | top5 % |
|---|---|---|---|---|
| XNOR-Net Rastegari et al. (2016) | ResNet-18 | 1-bit | 51.2 | 73.2 |
| Bi-Real Net Liu et al. (2018) | ResNet-18 | 1-bit | 56.4 | 79.5 |
| Bethge et al. (2019) | BinaryResNetE18 | 1-bit | 58.1 | 80.6 |
| Bethge et al. (2019) | BinaryDenseNet28 | 1-bit | 60.7 | 82.4 |
| **ours** | BinaryResNetE18 | 1-bit | 58.1 | 80.6 |
| **ours** | BinaryDenseNet28 | 1-bit | 60.7 | 82.4 |

Table 2: Accuracy comparison of our method with SOTA on ImageNet.

## 5   CONCLUSION

This paper introduced several optimization in the BNN data-flow that together achieve a speed-up of $1.77$ and $1.9\times$ compared to state-of-the-art BNNs frameworks, without any accuracy loss for at least one full-precision model. In the future, we intend to investigate: (i) the application of similar optimization techniques to ternary neworks, which naturally get higher accuracies; (ii) the simplification of the training procedure, possibly collapsing it to a single stage to further reduce training time and complexity.

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

## 6 BATCH NORMALIZATION QUANTIZATION PROCEDURE

The 8-bit quantization procedure used to quantize the BN layers, as shown in Fig. 3, is reported in Algorithm 2. The quantization procedure adopted is symmetric and keeps unaltered the zero point representation.

---

**Algorithm 2:** Procedure to quantize the BN floating point layers in a BNN model where convolution output is clipped.

---

**Input:** The full-precision weights W; the input training dataset
**Output:** BNN model with BN float layers replaced by 8-bit quantized version

**1** **for** $l = 1$ *to* $L$ **do**

**2**    **if** *l is BN floating point* **then**

**3**      Compute range of features $F^l_{out}$ as: $Range^l = \left[ min\left( F^l_{out} \right); max\left( F^l_{out} \right) \right]$

     // $l^N$ is the number of layer variables (4 for BN)

**4**      **for** $h = 1$ *to* $l^N$ **do**

**5**        Compute range of variable $w^l_h$ as: $Range_{w^l_h} = \left[ min\left( w^l_h \right); max\left( w^l_h \right) \right]$

       // 1 bit is reserved for sign

**6**        Compute number of bits used for range as: $RangeBits_{w^l_h} =$

       $clip\left( \left\lceil log_2\left( max\left( abs\left( Range_{w^l_h}[0] \right), \left( Range_{w^l_h}[1] \right) \right) \right) \right\rceil, 0, 15 \right)$

**7**        Compute number of bits used for fractional part as: $FracBits_{w^l_h} = 15 - RangeBits_{w^l_h}$

**8**      Select the Integer part (range) for all $N$ weights as: $RangeBits_{w^l} = max\left( RangeBits_{w^l_h} \right)$

**9**      Select the Fractional part for all weights as: $FracBits_{w^l} = 15 - RangeBits_{w^l}$

**10**      **for** $h = 1$ *to* $l^N$ **do**

**11**        Add quantization noise to floating point weights $w^l_h$ as: $w^l_{q_h} = \frac{1}{FracBits_{w^l}} round\left( 2^{FracBits_{w^l}} * w^l_h \right)$

**12**        Replace $w^l_h$ with $w^l_{q_h}$

**13**      Freeze $w^l$ weights and retrain the model

     // Export the quantized weights of layer $l$ for deployment

**14**      **for** $h = 1$ *to* $l^N$ **do**

**15**        $w^l_{quantized_h} = round\left( 2^{FracBits_{w^l}} * w^l_{q_h} \right)$

