# OpenReview forum: "Optimizing Data-Flow in Binary Neural Networks"
_ICLR.cc/2023/Conference — Submitted to ICLR 2023_

### Official Review · Reviewer_yPN2 · 2022-10-14

**Confidence:** 5
**Correctness:** 2
**Technical Novelty And Significance:** 1
**Empirical Novelty And Significance:** 1
**Recommendation:** 3

**Clarity, Quality, Novelty And Reproducibility:**

Overall the paper is easy to follow and most of the details are provided, although a code release would be recommended to allow for full reproducibility.

On somewhat of a technicality, Section 6 and algorithm 2 are not clearly delimitated as appendix.

**Strength And Weaknesses:**

Strengths:
- Good latency improvement on Raspberry Pi 3 and 4.
- Overall, the paper is easy to follow.

Weaknesses:
1. Limited novelty. It is widely accepted by the community that the main remaining bottleneck in the throughput of binary network is the presence of real valued operations and data. For example, [A] removes completly the BN, while [B] proposes a binary-friendly shift based implementation. While other works investigate quantizing to int8 the BNs and the first and last layer, with [C] investigating the impact of the data width both at train and test time.
The proposed modification are incremental at best and are of limited novelty.

2. Insufficient evaluation. Many of the recent models such as [D,E] but not limited to, make various architectural changes to the activation functions, learning various scaling factors and exotic layers. Significant testing is required with such networks that represent the current state of the art. In certain cases the gap between the tested models and current state of the art is more than 10% on ImageNet.

3. Insufficient related work. The related work section is incomplete and misses important works. Most of the methods (network and methodology wise) are before 2019.

4. There are results on a per-layer basis, but no aggregated ones per model. What is the speedup in this case?

5. Perhaps its just a strange coincidence, but why the results from Table 2 before and after re-training have exactly the same acc? I would expect the int8 quantization + 2 staged training to have perturbed the results in either directions.

6. 2-staged approaches are common for training binary networks. Where the baselines from Table 2 also trained using a stage approach for fairness?


[A] "BNN - BN = ?": Training Binary Neural Networks without Batch Normalization, Chen et al, CVPR-W'21
[B] Binarized Neural Networks: Training Neural Networks with Weights and Activations Constrained to +1 or −1, Courbariaux et al, 2016
[C] Enabling Binary Neural Network Training on the Edge, Wang et al 2021
[D] "ReActNet: Towards Precise Binary NeuralNetwork with Generalized Activation Functions", Liu et al, ECCV'20
[E] High-Capacity Expert Binary Networks, Bulat et al, ICLR'21

**Summary Of The Paper:**

The paper analyses the effect of the width of the data pipeline on the throughput of binary neural networks advocating for a decrease in the number of bits from 32 to 8. A series of adjustments are made to ensure, on the selected model that the performance doesn't drop when switching 32 to 8 bits (including a 2 staged training procedure). The paper also details an implementation technique for such binary convolutions on ARM CPUs. However, the proposed modifications are minor adjustments of existing techniques and are of limited novelty. Moreover, the architecture used are not representative of the current state of the art.

**Summary Of The Review:**

The main concerns with the paper are: insufficient/incremental novelty and insufficient empirical results, that do not reflect the current state of the art.

---

### Official Review · Reviewer_GiJD · 2022-10-25

**Confidence:** 4
**Correctness:** 3
**Technical Novelty And Significance:** 1
**Empirical Novelty And Significance:** 1
**Recommendation:** 3

**Clarity, Quality, Novelty And Reproducibility:**

The paper provides a set of engineering hacks to reduce the overhead of BNN execution on ARM CPUs, however, the proposed techniques are not particularly novel. There are no new technical methods introduced for reducing the overhead. Please see the weaknesses above.

**Strength And Weaknesses:**

Strengths:
- The paper is well-written and it is easy to understand the methodologies applied.

Weaknesses:
- The paper has very limited novelty. Reducing the intermediate bitwidths to 8 bits rather than the original full-precision values is a technique that has been performed for many years in academia and industry to reduce the overheads, and the particular gradual mapping of 32-bit values to 8-bits during training is also not particularly novel. Additionally, the conversion of full-precision BN operations to 8-bit ones is a natural extension of having 8-bit input/output operands and is not a new contribution on its own.
- The experimental results only show speedups on specific example layers (see Figure 6). But the important question is what are the "end-to-end" latency measurement for a "full network" and how does that compare with prior art?

**Summary Of The Paper:**

This paper proposes some modifications to the execution flow of BNNs for better performance on low-end hardware, specifically ARM CPUs. Firstly, the authors reduce 32-bit intermediate representations after the XNOR popcount in the BNN to 8-bit values to reduce overhead. Secondly, they replace BN with a comparison operation or use 8-bit quantized BN statistics to perform the computations in lower bitwidth arithmetic. The authors also use specific ARM instructions to implement the BNNs more efficiently.

**Summary Of The Review:**

Please see the weaknesses above.

---

### Official Review · Reviewer_mSLQ · 2022-10-26

**Confidence:** 4
**Correctness:** 3
**Technical Novelty And Significance:** 2
**Empirical Novelty And Significance:** 2
**Recommendation:** 3

**Clarity, Quality, Novelty And Reproducibility:**


The provided methods and solutions are easy to understand.


**Strength And Weaknesses:**

Weaknesses: The paper lacks novelty for a conference like ICLR. Quantization of BN into 8-bits has already been proposed in previous works and the simplification of BN is obvious. The impact of their implementation of BN layer on the acceleration of BNN is not explored.

Strength: The proposed assembly implementation of the binary convolution.

**Summary Of The Paper:**

The paper tries to accelerate the inference of BNNs by:

1- Quantizing the BatchNorm layer into 8-bits
2- Simplifying the deployment of BN layer non ResNet like networks
3- And proposing an optimized assembly implementation of the binary convolution

**Summary Of The Review:**

Due to the lack of novelty I recommend to reject the paper.

---

### Official Review · Reviewer_vs6q · 2022-10-26

**Confidence:** 4
**Correctness:** 2
**Technical Novelty And Significance:** 1
**Empirical Novelty And Significance:** 3
**Recommendation:** 3

**Clarity, Quality, Novelty And Reproducibility:**

I believe the paper falls short in terms of novelty (see my comments listed as weaknesses). The contribution of this easy to understand but there are some aspects missing such as scaling factor of binarization.


**Strength And Weaknesses:**

Strengths:
-- The paper provides an insightful challenges about hardware implementation challenges of binary neural networks on ARM.
-- It provides a hardware implementation on ARM and yields up to 1.9x speedup compared to prior works.
-- The paper is well-written and easy to understand.

Weaknesses:
-- The two-step clipping method was introduced to accommodate for 8-bit batch normalization. However, it doesn't seem to be necessary for VGG-style block. According to Eq. (2), all we need to compute is sign of the batch norm's output, which can be achieved by comparing the input of batch norm with the threshold tau. Of course, tau can be represented using 8 bits. Then, the input of batch norm doesn't need to be represented using 32 bits anymore; its 8-bit representation suffices. I am also assuming the scaling factor of binarization is merged with BN, is it a right assumption? For the ResNet-style block, the batch norm is mixed-precision not 8-bit according to Fig. 3. Then, what's the point of representing the input using 8 bits?

-- 8-bit batch norm has been proposed before in literature (e.g., https://arxiv.org/pdf/1805.11046.pdf), which undermines the novelty of this work.

**Summary Of The Paper:**

This paper presents a training scheme to increase data pipeline of binary neural networks when implemented on ARM. To this end, a two-step clipping method and 8-bit batch normalization have been proposed. The clipping method insures that the values can fit into 8 bits so that batch normalization can be performed using the same bitwidth. The proposed method has no impact on accuracy performance and its implementation on Raspberry Pi shows up to 1.9x speedup compared to prior works.


**Summary Of The Review:**

I believe this paper suffers from the lack of motivation and also novelty. The statements of this paper don't explain why two-step clipping is required to make the inputs of BN into 8 bits while BN is in mixed-precision (8-bit and 16-bit). Besides, 8-bit batch norm has been introduced before in different works.

---

### Decision · Program_Chairs · 2023-01-20

**Decision:**

Reject

**Justification For Why Not Higher Score:**

low scores, no rebuttal, lacking a substantial contribution

**Justification For Why Not Lower Score:**

N/A

**Metareview: Summary, Strengths And Weaknesses:**

### Decision

The paper received very low initial scores. The main common point of criticism is that many efficiency improvements would be known to a skilled software engineer. As such there is no sufficient contribution for ICLR. Since there was also no rebuttal, I did not see it necessary to go into details --- the paper is to be rejected.